# AMR Parsing with Causal Hierarchical Attention and Pointers

**Chao Lou, Kewei Tu**[*]
School of Information Science and Technology, ShanghaiTech University
Shanghai Engineering Research Center of Intelligent Vision and Imaging
{louchao,tukw}@shanghaitech.edu.cn

## Abstract

Translation-based AMR parsers have recently gained popularity due to their simplicity and effectiveness. They predict linearized graphs as free texts, avoiding explicit structure modeling. However, this simplicity neglects structural locality in AMR graphs and introduces unnecessary tokens to represent coreferences. In this paper, we introduce new target forms of AMR parsing and a novel model, CHAP, which is equipped with **c**ausal **h**ierarchical **a**ttention and the **p**ointer mechanism, enabling the integration of structures into the Transformer decoder. We empirically explore various alternative modeling options. Experiments show that our model outperforms baseline models on four out of five benchmarks in the setting of no additional data.

## 1 Introduction

Abstract Meaning Representation (Banarescu et al., 2013) is a semantic representation of natural language sentences typically depicted as directed acyclic graphs, as illustrated in Fig. 1a. This representation is both readable and broad-coverage, attracting considerable research attention across various domains, including information extraction (Zhang and Ji, 2021; Xu et al., 2022), summarization (Hardy and Vlachos, 2018; Liao et al., 2018), and vision-language understanding (Schuster et al., 2015; Choi et al., 2022). However, the inherent flexibility of graph structures makes AMR parsing , i.e., translating natural language sentences into AMR graphs, a challenging task.

The development of AMR parsers has been boosted by recent research on pretrained sequence-to-sequence (seq2seq) models. Several studies, categorized as translation-based models, show that fine-tuning pretrained seq2seq models to predict linearized graphs as if they are free texts (e.g., examples in Tab.1.ab) can achieve competitive or

even superior performance (Konstas et al., 2017; Xu et al., 2020; Bevilacqua et al., 2021; Lee et al., 2023). This finding has spurred a wave of subsequent efforts to design more effective training strategies that maximize the potential of pretrained decoders (Bai et al., 2022; Cheng et al., 2022; Wang et al., 2022; Chen et al., 2022), thereby sidelining the exploration of more suitable decoders for graph generation. Contrary to preceding translation-based models, we contend that explicit structure modeling within pretrained decoders remains beneficial in AMR parsing. To our knowledge, the Ancestor parser (Yu and Gildea, 2022) is the only translation-based model contributing to explicit structure modeling, which introduces shortcuts to access ancestors in the graph. However, AMR graphs contain more information than just ancestors, such as siblings and coreferences, resulting in suboptimal modeling.

In this paper, we propose CHAP, a novel translation-based AMR parser distinguished by three innovations. Firstly, we introduce new target forms of AMR parsing. As demonstrated in Tab. 1.c-e, we use multiple layers to capture different semantics, such that each layer is simple and concise. Particularly, the base layer, which encapsulates all meanings except for coreferences (or reentrancies), is a tree-structured representation, enabling more convenient structure modeling than the graph structure of AMR. Meanwhile, coreferences are presented through pointers, circumventing several shortcomings associated with the variable-based coreference representation (See Sec. 3 for more details) used in all previous translation-based models. Secondly, we propose Causal Hierarchical Attention (CHA), the core mechanism of our incremental structure modeling, inspired by Transformer Grammars (Sartran et al., 2022). CHA describes a procedure of continuously composing child nodes to their parent nodes and encoding new nodes with all uncomposed nodes, as illustrated in Fig. 2. Un-

---

[*]Corresponding Author

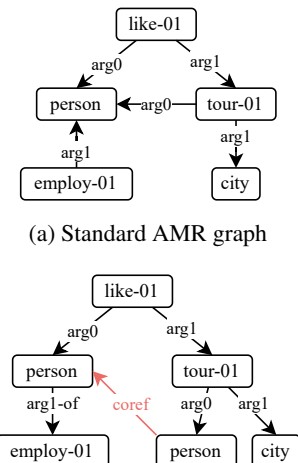

(a) Standard AMR graph

(b) Replicate nodes and add coref pointers (red).

```
( l / like-01 :arg0 ( p
/ person :arg1-of ( e /
employ-01 ) ) :arg1 ( t
/ tour-01 :arg0 p :arg1
( c / city ) ) )
```

(c) PENMAN notation

Figure 1: AMR of *Employees liked their city tour.*

| ID | Name | Representation |
|---|---|---|
| a | PM | ( a / alpha :arg0 ( b / beta ) :arg1 ( g / gamma :arg2 b ) ) |
| b | S-DFS | ( <R0> alpha :arg0 ( <R1> beta ) :arg1 ( <R2> gamma :arg2 <R1> ) ) |
| c | ⇓ sgl. | ( alpha :arg0 ( beta ) :arg1 ( gamma :arg2 beta ) ) |
| d | ⇓ dbl. | ( alpha :arg0 ( beta )₁ )₂ :arg1 ( gamma :arg2 beta )₁ )₂ )₁ )₂ |
| e | ⇑ | alpha :arg0 beta ■ :arg1 gamma :arg2 beta ■ ■ |

Table 1: Graph representations. PM and S-DFS denote the PENMAN form and the SPRING_DFS form, the DFS-based linearization proposed by Bevilacqua et al. (2021), respectively. (c)-(d) are our proposed representations. (c) is for ⇓single(c) (Fig. 3c). (d) is for ⇓double(c) (Fig. 3b). (e) is for ⇑ (Fig. 3d). Red pointers ⤸, which represent coreferences, constitute the coref layer. Blue pointers↑, which point to the left boundaries of subtrees, constitute the auxiliary structure layer. Texts, which encapsulate all other meanings, constitute the base layer.

| Property | Translation-based | Transition-based | Factor.-based | Ours |
|---|---|---|---|---|
| Trainability | ✓ | require alignments | ✓ | ✓ |
| Structure modeling | ✗ | ✗ | ✓ | ✓ |
| Pretrained decoder | ✓ | ✓ | ✗ | ✓ |
| Variable-free | ✗ | ✓ | ✓ | ✓ |

Table 2: Strengths, shortcomings of different types of models.

like the causal attention in translation-based models, which allows a token to interact with all its preceding tokens, CHA incorporates a strong inductive bias of recursion, composition, and graph topology. Thirdly, deriving from transition-based AMR parsers (Zhou et al., 2021a,b), we introduce a pointer encoder for encoding histories and a pointer net for predicting coreferences, which is proven to be an effective solution for generalizing to a variable-size output space (Vinyals et al., 2015; See et al., 2017).

We propose various alternative modeling options of CHA and strategies for integrating CHA with existing pretrained seq2seq models and investigate them via extensive experiments. Ultimately, our model CHAP achieves superior performance on two in-distribution and three out-of-distribution benchmarks. Our code is available at https://github.com/LouChao98/chap_amr_parser.

## 2 Related Work

### 2.1 AMR Parsing

Most recent AMR parsing models generate AMR graphs via a series of local decisions. Transition-based models (Ballesteros and Al-Onaizan, 2017; Naseem et al., 2019; Fernandez Astudillo et al.,

2020; Zhou et al., 2021a,b) and translation-based models (Konstas et al., 2017; Xu et al., 2020; Bevilacqua et al., 2021; Lee et al., 2023) epitomize local models as they are trained with teacher forcing, optimizing only next-step predictions, and rely on greedy decoding algorithms, such as greedy search and beam search. Particularly, transition-based models predict actions permitted by a transition system, while translation-based models predict AMR graph tokens as free texts. Some factorization-based models are also local (Cai and Lam, 2019, 2020), sequentially composing subgraphs into bigger ones. We discern differences in four properties among previous local models and our model in Tab. 2:

**Trainability**  Whether additional information is required for training. Transition-based models rely on word-node alignment to define the gold action sequence.

**Structure modeling**  Whether structures are modeled explicitly in the decoder. Transition-based models encode action histories like texts without considering graph structures. Besides, translation-based models opt for compatibility with pretrained decoders, prioritizing this over explicit structure

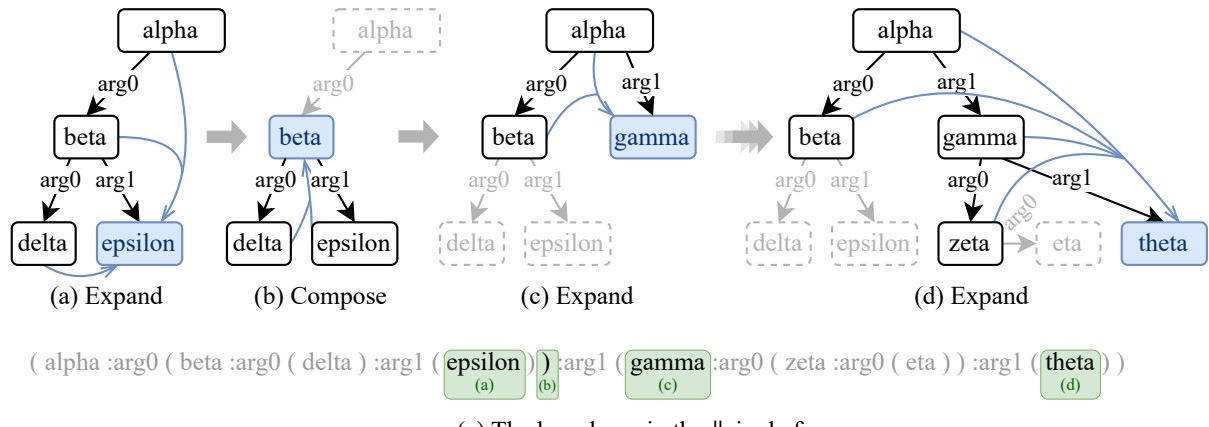

(a) Expand     (b) Compose     (c) Expand     (d) Expand

( alpha :arg0 ( beta :arg0 ( delta ) :arg1 ( epsilon ) ) :arg1 ( gamma :arg0 ( zeta :arg0 ( eta ) ) :arg1 ( theta ) )

(e) The base layer in the ⇓single form.

Figure 2: Demonstration of Causal Hierarchical Attention. We draw the aggregation on graphs performed at four steps in (a)-(d) and highlight the corresponding token for each step in (e) with green boxes. the The generation order is depth-first and left-to-right: alpha → beta → delta → epsilon → gamma → zeta → eta → theta. The node of interest at each step is highlighted in blue, gathering information from all solid nodes. Gray dashed nodes, on the other hand, are invisible.

modeling.

**Pretrained decoder** Whether pretrained decoders can be leveraged.

**Variable-free** Whether there are variable tokens in the target representation. Transition-based models, factorization-based models and ours generate coreference pointers, obviating the need of introducing variables.

### 2.2 Transformer Grammar

Transformer Grammars (TGs; Sartran et al., 2022) are a novel class of language models that simultaneously generate sentences and constituency parse trees, in the fashion of transition-based parsing. The base layer of Tab. 1.c can be viewed as an example action sequence. There are three types of actions in TG: (1) the token "(" represents the action ONT, opening a nonterminal; (2) the token ")" represents the action CNT, closing the nearest open nonterminal; and (3) all other tokens (e.g., a and :arg0) represent the action T, generating a terminal. TG carries out top-down generation, where a nonterminal is allocated before its children. We will also explore a bottom-up variant in Sec. 3.4. Several studies have already attempted to generate syntax-augmented sequences (Aharoni and Goldberg, 2017; Qian et al., 2021). However, TG differentiates itself from prior research through its unique simulation of stack operations in transition-based parsing, which is implemented by enforcing a specific instance of CHA. A TG-like CHA is referred

to as ⇓double in this paper and we will present technical details in Sec. 3.3 along with other variants.

## 3 Structure Modeling

We primarily highlight two advantages of incorporating structured modeling into the decoder. Firstly, the sequential order and adjacence of previous linearized form mismatch the locality of real graph structures, making the Transformer decoder hard to understand graph data. Specifically, adjacent nodes in an AMR graph exhibit strong semantic relationships, but they could be distant in the linearized form (e.g., person and tour-01 in Fig. 1a). Conversely, tokens closely positioned in the linearized form may be far apart in the AMR graph (e.g., employ-01 and tour-01 in Fig. 1a). Secondly, previous models embed variables into the linearized form (e.g., b in Tab. 1.a and <R1> in Tab. 1.b) and represent coreferences (or reentrancies) by reusing the same variables. However, the literal value of variables is inconsequential. For example, in the PENMAN form, (a / alpha :arg0 (b / beta)) conveys the same meaning as (n1 / alpha :arg0 (n2 / beta)). Furthermore, the usage of variables brings up problems regarding generalization (Wong and Mooney, 2007; Poelman et al., 2022). For instance, in the SPRING_DFS form, <R0> invariably comes first and appears in all training samples, while <R100> is considerably less frequent.

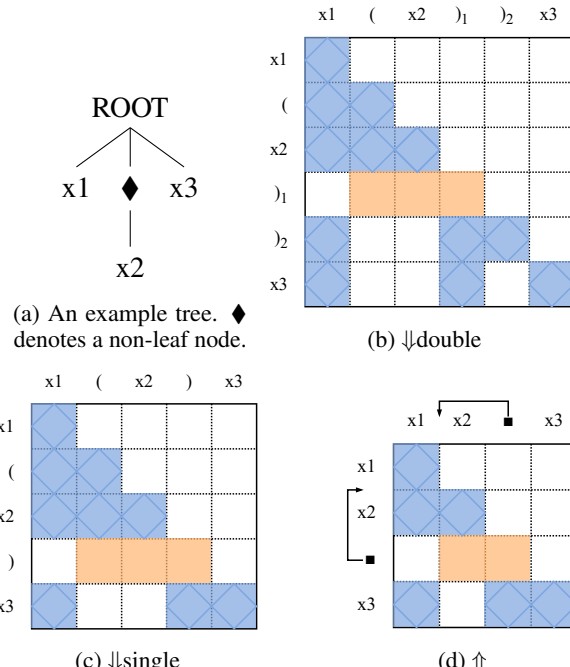

(a) An example tree. ♦ denotes a non-leaf node.

(b) ⇓double

(c) ⇓single

(d) ⇑

Figure 3: The target forms and the attention mask of the three variants of CHA for the tree in (a). Orange cells represent the *compose* operation, while blue cells with plaid represent the *expand* operation. White cells are masked out. The vertical and horizontal axis represent attending and attended tokens, respectively.

## 3.1 Multi-layer Target Form

As shown in Tab. 1.c, we incorporate a new layer, named the coreference (coref) layer, on top of the conventional one produced by a DFS linearization, named the base layer[1]. The coref layer serves to represent coreferences, in which a pointer points from a mention to its nearest preceding mention, and the base layer encapsulates all other meanings. From a graph perspective, a referent is replicated to new nodes with an amount equal to the reference count that are linked by newly introduced coref pointers, as illustrated in Fig. 1b. We argue that our forms are more promising because our forms can avoid meaningless tokens (i.e., variables) from cluttering up the base layer, yielding several beneficial byproducts: (1) it shortens the representation length; (2) it aligns the representation more closely with natural language; and (3) it allows the base layer to be interpreted as trees, a vital characteristic for our structure modeling.

Tab. 1.d and e are two variants of Tab. 1.c. These three forms are designed to support different variants of CHA, which will be introduced in Sec. 3.3

[1]We use the DFS order provided in the AMR datasets.

and 3.4.

## 3.2 Causal Hierarchical Attention

Causal Hierarchical Attention (CHA) is situated in the decoder and maintains structures during generation. For each token, CHA performs one of the two actions, namely *compose* and *expand*, as demonstrated in Fig. 2. The *compose* operation is performed once all children of a parent node have been generated. It aggregates these children to obtain a comprehensive representation of the subtree under the parent node, subsequently setting the children invisible in future attention. On the other hand, the *expand* operation aggregates all visible tokens to derive the representation of the subsequent token.

We note the subtle distinction between the term *parent* in the DAG representation (e.g., Fig. 1a) and in target forms (e.g., Tab. 1 and Fig. 3a). Recall that TG uses "(" (ONT) to indicate a new nonterminal, which is the parent node of all following tokens before a matched ")" (CNT). This implies that alpha in Tab. 1.c is considered as a child node, being a sibling of :arg0, rather than a parent node governing them. This discrepancy does not impact our modeling because we can treat labels of nonterminals as a particular type of child nodes, which are absorbed into the parent node when drawing the DAG representation. The tree of target forms are illustrated in Appx. A.1.

The two actions can be implemented by modifying attention masks conveniently. Specially, the *compose* operation masks out attention to tokens that are not destined to be composed, as depicted in the fourth row of Fig. 3c. Moreover, the *expand* operation masks out attention to tokens that have been composed in previous steps, as depicted in the top three rows and the fifth row of Fig. 3c.

In subsequent sections, we will explore two classes of generation procedures that utilize different target forms and variants of CHA, akin to top-down (Dyer et al., 2016; Nguyen et al., 2021; Sartran et al., 2022) and bottom-up (Yang and Tu, 2022) parsing.

## 3.3 Top-down generation

Most prior studies utilize paired brackets to denote nested structures, as shown in Tab. 1.a-d. Section 2.2 outlines that, in left-to-right decoding, due to the prefix "(", this type of representation results in top-down tree generation.

We consider two modeling options of top-down generation, ⇓single (Fig. 3c and ⇓double (Fig. 3b), varying on actions triggered by ")". More precisely, upon seeing a ")", ⇓single executes a *compose*, whereas ⇓double executes an additional *expand* after the *compose*. For other tokens, both ⇓single and ⇓double execute an *expand* operation. Because the decoder performs one attention for each token, in ⇓double, each ")" is duplicated to represent *compose* and *expand* respectively, i.e., ")" becomes ")$_1$ )$_2$". We detail the procedure of generating $M_{\text{CHA}}$ for ⇓single in Alg. 1. The procedure for ⇓double can be found in Sartran et al.'s (2022) Alg. 1.

The motivation for the two variants is as follows. In a strict leaf-to-root information aggregation procedure, which is adopted in many studies on tree encoding (Tai et al., 2015; Drozdov et al., 2019; Hu et al., 2021; Zhou et al., 2022), a parent node only aggregates information from its children, remaining unaware of other generated structures (e.g., beta is unaware of alpha in Fig. 2b). However, when new nodes are being expanded, utilizing all available information could be a more reasonable approach (e.g., gamma in Fig. 2c). Thus, an *expand* process is introduced to handle this task. The situation with CHA becomes more flexible. Recall that all child nodes are encoded with the *expand* action, which aggregates information from all visible nodes, such that information of non-child nodes is leaked to the parent node during composition. ⇓single relies on the neural network's capability to encode all necessary information through this leakage, while ⇓double employs an explicit *expand* to allow models to directly revisit their histories.

### 3.4 Bottom-up generation

In the bottom-up generation, the parent node is allocated after all child nodes have been generated. This process enables the model to review all yet-to-be-composed tokens before deciding which ones should be composed into a subtree, in contrast to the top-down generation, where the model is required to predict the existence of a parent node without seeing its children. The corresponding target form, as illustrated in Tab. 1.e, contains no brackets. Instead, a special token ■ is placed after the rightmost child node of each parent node, with a pointer pointing to the leftmost child node. We execute the *compose* operation for ■ and the *expand* operation for other tokens. The generation of the attention mask (Fig. 3d) is analogous to ⇓single,

---

**Algorithm 1:** $M_{\text{CHA}}$ for ⇓single.

**Data:** sequence of token $t$ with length $N$
**Result:** attention mask $M_{\text{CHA}} \in \mathbb{R}^{N \times N}$
$S \leftarrow [\,]$             ▷ Empty stack
$M_{\text{CHA}} \leftarrow -\infty$
**for** $i \leftarrow 1$ **to** $N$ **do**
   **if** $t[i] = $ ')' **then**       ▷ compose
      $j \leftarrow i$
      **while** $t[j] \neq $ '(' **do**
         $M_{\text{CHA}}[ij] \leftarrow 0$
         $j \leftarrow S.pop()$
      **end**
      $M_{\text{CHA}}[ij] \leftarrow 0$
      $S.push(i)$
   **else**
      $S.push(i)$
      **for** $j \in S$ **do**       ▷ expand
         $M_{\text{CHA}}[ij] \leftarrow 0$
      **end**
   **end**
**end**
**return** $M_{\text{CHA}}$

---

but we utilize pointers in place of left brackets to determine the left boundaries of subtrees. The exact procedure can be found in Appx. B.

## 4 Parsing Model

Our parser is based on BART (Lewis et al., 2020), a pretrained seq2seq model. We make three modifications to BART: (1) we add a new module in the decoder to encode generated pointers, (2) we enhance decoder layers with CHA, and (3) we use the pointer net to predict pointers.

### 4.1 Encoding Pointers

The target form can be represented as a tuple of $(t, p)$[2], where $t$ and $p$ are the sequence of the base layer and the coref layer, respectively, such that each $p_i$ is the index of the pointed token. We define $p_i = -1$ if there is no pointer at index $i$.

In the BART model, $t$ is encoded using the token embedding. However, no suitable module exists for encoding $p$. To address this issue, we introduce a multi-layer perceptron, denoted as $\text{MLP}_p$, which takes in the token and position embeddings of the pointed tokens and then outputs the embedding

---

[2]For the sake of simplicity, we only discuss the target form of top-down generation. The additional struct layer in the bottom-up generation can be modeled similarly to $p$.

of $p$. Notably, if $p_i = -1$, the embedding is set to 0. All embeddings, including that of $t$, $p$ and positions, are added together before being fed into subsequential modules.

## 4.2 Augmenting Decoder with CHA

We explore three ways to integrate CHA in the decoder layer, as shown in Fig. 4. The *inplace* architecture replaces the attention mask of some attention heads with $M_{\text{CHA}}$ in the original self-attention module without introducing new parameters. However, this affects the normal functioning of the replaced heads such that the pretrained model is disrupted.

Alternatively, we can introduce adapters into decoder layers (Houlsby et al., 2019). In the *parallel* architecture, an adapter is introduced in parallel to the original self-attention module. In contrast, an adapter is positioned subsequent to the original module in the *pipeline* architecture. Our adapter is defined as follows:

$$x_1 = \text{FFN}_1(h_i),$$
$$x_2 = \text{Attention}(W^Q x_1, W^K x_1, W^V x_1, M_{\text{CHA}}),$$
$$h_o = \text{FFN}_2(\text{LayerNorm}(x_1 + x_2)),$$

where $W^Q, W^K, W_V$ are query/key/value projection matrices, $\text{FFN}_1/\text{FFN}_2$ are down/up projection, $h_i$ is the input hidden states and $h_o$ is the output hidden states.

## 4.3 Predicting Pointer

Following previous work (Vinyals et al., 2015; Zhou et al., 2021b), we reinterpret decoder self-attention heads as a pointer net. However, unlike the previous work, we use the average attention probabilities from multiple heads as the pointer probabilities instead of relying on a single head. Our preliminary experiments indicate that this modification results in a slight improvement.

A cross-entropy loss between the predicted pointer probabilities and the ground truth pointers is used for training. We disregard the associated loss at positions that do not have pointers and exclude their probabilities when calculating the entire pointer sequence's probability.

## 4.4 Training and Inference

We optimize the sum of the standard sequence generation loss and the pointer loss:

$$L = L_{\text{seq2seq}} + \alpha L_{\text{pointer}},$$

where $\alpha$ is a scalar hyperparameter.

For decoding, the probability of a hypothesis is the product of the probabilities of the base layer, the coref sequence, and the optional struct layer. We enforce a constraint during decoding to ensure the validity of $M_{CHA}$: the number of ) should not surpass the number of (, and two constraints to ensure the well-formedness of pointer: (1) coreference pointers can only point to positions with the same token, and (2) left boundary pointers in bottom-up generation cannot point to AMR relations (e.g., :ARG0).

# 5 Experiment

## 5.1 Setup

**Datasets** We conduct experiments on two in-distribution benchmarks: (1) AMR 2.0 (Knight et al., 2017), which contains 36521, 1368 and 1371 samples in the training, development and test set, and (2) AMR 3.0 (Knight et al., 2020), which has 55635, 1722 and 1898 samples in the training, development and test set, as well as three out-of-distribution benchmarks: (1) *The Little Prince* (TLP), (2) BioAMR and (3) New3. Besides, we also explore the effects of using silver training data following previous work. To obtain silver data, we sample 200k sentences from the One Billion Word Benchmark data (Chelba et al., 2014) and use a trained CHAP parser to annotate AMR graphs.

**Metrics** We report the Smatch score (Cai and Knight, 2013) and other fine-grained metrics (Damonte et al., 2017) averaged over three runs with different random seeds[3]. All these metrics are invariant to different graph linearizations and exhibit better performance when they are higher. Additionally, to provide a more accurate comparison, we include the standard deviation (std dev) if Smatch scores are close.

**Pre-/post-processing** Owing to the sparsity of wiki tags[4] in the training set, we follow previous work to remove wiki tags from AMR graphs in the pre-processing, and use the BLINK entity linker (Wu et al., 2020) to add wiki tags in the post-processing[5]. In the post-processing, we also use

---

[3]We use the amr-evaluation-enhanced software to compute scores, which is available at https://github.com/ChunchuanLv/amr-evaluation-tool-enhanced.

[4]https://github.com/amrisi/amr-guidelines/blob/master/amr.md#named-entities

[5]We do not add wiki tags in analytical experiments.

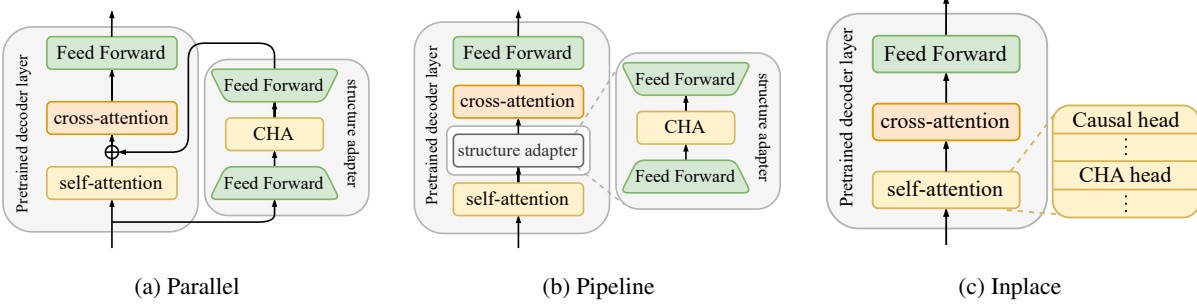

|                | (a) Parallel | (b) Pipeline | (c) Inplace |

Figure 4: Three architectures for applying CHA to pretrained decoder layers. Residual connections and layernorms are omitted.

the `amrlib` software[6] to ensure graph validity.

**Implementation details**  We use the BART-base model in analytical experiments and the BART-large model in comparison with baselines. We modify all decoder layers when using the BART-base model, while only modifying the top two layers when using the BART-large model[7]. For the parallel and pipeline architectures, attention modules in adapters have four heads and a hidden size 512. For the inplace architecture, four attention heads are set to perform CHA. We reinterpret four self-attention heads of the top decoder layer as a pointer net. The weight for the pointer loss $\alpha$ is set to $0.075$. We use a zero initialization for $FFN_2$ and $MLP_p$, such that the modified models are equivalent to the original BART model at the beginning of training. More details are available at Appx. D

**Baselines**  SPRING (Bevilacqua et al., 2021) is a BART model fine-tuned with an augmented vocabulary and improved graph representations (as shown in Tab. 1.b). Ancestor (Yu and Gildea, 2022) enhances the decoder of SPRING by incorporating ancestral information of graph nodes. BiBL (Cheng et al., 2022) and AMRBART (Bai et al., 2022) augment SPRING with supplementary training losses. LeakDistill (Vasylenko et al., 2023)[8] trains a SPRING using leaked information and then distills it into a standard SPRING.

All these baselines are translation-based models. Transition-based and factorization-based models are not included due to their inferior performance.

| CHA | Smatch |
|---|---|
| ⇓single | 82.60 |
| expand → causal | 82.53 |
| compose → expand | 82.47 |
| ⇓double | 82.63 |
| ⇑ | 82.57 |

Table 3: The influence of different CHA.

| Architecture | Smatch | std dev |
|---|---|---|
| Parallel | 82.63 | 0.02 |
| Pipeline | 82.59 | 0.05 |
| Inplace | 82.43 | 0.04 |
| w/o CHA | 82.38 | 0.12 |

Table 4: The influence of different architectures.

### 5.2 Results on Alternative Modeling Options

**Structural modeling**  We report the results of different CHA options in Tab. 3. ⇓double exhibits a slightly better performance than ⇑ and ⇓single. Besides, we find that breaking structural localities, i.e., (1) allowing parent nodes to attend to nodes other than their immediate children (row 3, $-0.13$) and (2) allowing non-parent nodes to attend to nodes that have been composed (row 2, $-0.07$), negatively impacts the performance. We present the attention masks of these two cases in Appx. A.2.

**Architecture**  In Tab. 4, we can see that the inplace architecture has little improvement over the baseline, w/o CHA. This suggests that changing the functions of pretrained heads can be harmful. We also observe that the parallel architecture performs slightly better than the pipeline architecture.

Based on the above results, we present CHAP, which adopts the parallel adapter and uses ⇓double.

---

[6] https://github.com/bjascob/amrlib
[7] The training becomes unstable if we modify all decoder layers of the BART-large model.
[8] Contemporary work.

| Model | Extra Data | Smatch | NoWSD | Wiki. | Conc. | NER | Neg. | Unlab | Reent. | SRL |
|---|---|---|---|---|---|---|---|---|---|---|
| *AMR 2.0* | | | | | | | | | | |
| SPRING | – | 83.8 | 84.4 | 84.3 | 90.2 | 90.6 | 74.4 | 86.1 | 70.8 | 79.6 |
| Ancestor | – | 84.8 | 85.3 | 84.1 | 90.5 | 91.8 | 74.0 | 88.1 | **75.1** | **83.4** |
| BiBL | – | 84.6 | 85.1 | 83.6 | 90.3 | **92.5** | 73.9 | 87.8 | 74.4 | 83.1 |
| LeakDistill | A | **85.7** | **86.2** | 83.9 | **91.0** | 91.1 | **76.8** | 88.6 | 74.2 | 81.8 |
| CHAP (ours) | – | 85.1 | 85.6 | **86.4** | 90.9 | 90.4 | 73.4 | 88.0 | 73.0 | 81.0 |
| AMRBART | 200K | 85.4 | 85.8 | 81.4 | 91.2 | 91.5 | 74.0 | 88.3 | 73.5 | 81.5 |
| LeakDistill | A, 140K | **86.1** | **86.5** | 83.9 | **91.4** | **91.6** | 76.6 | **88.8** | **75.1** | **82.4** |
| CHAP (ours) | 200K | 85.8 | 86.1 | **86.3** | 91.4 | 80.4 | **78.3** | 88.6 | 73.9 | 81.8 |
| *AMR 3.0* | | | | | | | | | | |
| SPRING | – | 83.0 | 83.5 | 82.7 | 89.8 | 87.2 | 73.0 | 85.4 | 70.4 | 78.9 |
| Ancestor | – | 83.5 | 84.0 | 81.5 | 89.5 | 88.9 | 72.6 | 86.6 | **74.2** | **82.2** |
| BiBL | – | 83.9 | 84.3 | 83.7 | 89.8 | **93.2** | 68.1 | 87.2 | 73.8 | 81.9 |
| LeakDistill | A | **84.5** | **84.9** | 80.7 | **90.5** | 88.5 | **73.7** | 87.5 | 73.1 | 80.7 |
| CHAP (ours) | – | 84.4* | 84.8 | **84.7** | 90.5 | 87.9 | 73.5 | 87.3 | 72.6 | 80.1 |
| AMRBART | 200K | 84.2 | 84.6 | 78.9 | 90.2 | **88.5** | 72.1 | 87.1 | 72.4 | 80.3 |
| LeakDistill | A, 140K | **84.6** | 84.9 | 81.3 | **90.7** | 87.8 | 73.0 | **87.5** | **73.4** | **80.9** |
| CHAP (ours) | 200K | **84.6** | **85.0** | **84.5** | 90.7 | 88.4 | **75.2** | **87.5** | 73.1 | 80.7 |

Table 5: Fine-grained Smatch scores on in-domain benchmarks. Bold and underlined numbers represent the best and the second-best results, respectively. "A" in the Extra Data column denotes alignment. *Std dev is 0.04.

| Model | Extra Data | TLP | Bio | New3 |
|---|---|---|---|---|
| SPRING | – | 77.3 | 59.7 | 73.7 |
| BiBL | – | 78.6 | 61.0 | 75.4 |
| BiBL | 200K | 78.3 | 61.1 | 75.4 |
| AMRBART | 200K | 76.9 | 63.2 | 76.9 |
| LeakDistill | A, 140K | 82.6 | 64.5 | – |
| CHAP (ours) | – | 79.0 | 62.7 | 74.8 |
| CHAP (ours) | 200K | 79.8 | 63.5 | 75.1 |
| CHAP (ours) | – | 81.8 | 65.1 | – |
| CHAP (ours) | 200K | 82.7[α] | 66.1[β] | – |

Table 6: Test results on out-of-distribution benchmarks. The scores represented in grey cells derive from a model trained on AMR 2.0, whereas the remaining scores come from a model trained on AMR 3.0. −: New3 is part of AMR 3.0, so these settings are excluded from OOD evaluation. [α]Std dev on TLP is 0.14. [β]Std dev on Bio is 0.46.

## 5.3 Main Results

Tab. 5 shows results on in-distribution benchmarks. In the setting of no additional data (such that LeakDistill is excluded), CHAP outperforms all previous models by a 0.3 Smatch score on AMR 2.0 and 0.5 on AMR 3.0. Regarding fine-grained metrics, CHAP performs best on five metrics for AMR 3.0 and three for AMR 2.0. Compared to previous work, which uses alignment, CHAP matches LeakDistill on AMR 3.0 but falls behind it on AMR 2.0. One possible reason is that alignment as additional data is particularly valuable for a relatively small training set of AMR 2.0. We note that the contribution of LeakDistill is orthogonal to ours, and we can expect an enhanced performance by integrating their method with our parser. When using silver data, the performance of CHAP on AMR 2.0 can be significantly improved, achieving similar performance to LeakDistill. This result supports the above conjecture. However, on AMR 3.0, the gain from silver data is marginal as in previous work, possibly because AMR 3.0 is sufficiently large to train a model based on BART-large.

In out-of-distribution evaluation, CHAP is competitive with all baselines on both TLP and Bio, as shown in Tab. 6, indicating CHAP's strong ability of generalization thanks to the explicit structure modeling.

## 5.4 Ablation Study

An ablation study is presented in Table 7. The first four rows demonstrate that, for a model based on BART-base, when we exclude the encoding of pointers, the CHA adapters, and the separation

| EP | Adapter | CL | Base | Large | | |
|----|---------|----|------|-------|------|------|
| | | | AMR3 | AMR3 | TLP | Bio |
| ✓ | CHA | ✓ | **82.91** | **84.44** | 81.8 | **65.1** |
| ✗ | CHA | ✓ | 82.82 | 84.17 | 81.7 | 64.8 |
| ✓ | ✗ | ✓ | 82.75 | 84.28 | – | – |
| – | CHA | ✗ | 82.63 | 84.09 | **82.1** | 64.6 |
| – | Causal | ✗ | 82.44 | 83.79 | – | – |
| – | ✗ | ✗ | 82.38 | 83.84 | 81.6 | 64.3 |

Table 7: Ablation study. EP: Encoding Pointers. CL: Coreference Layer. Base(Large): BART-base(large).

of coreferences on AMR 3.0, there are declines in Smatch scores of $-0.09$, $-0.16$, and $-0.28$, respectively and $-0.27$, $-0.16$, and $-0.35$ for a model based on BART-large. Additionally, we substitute CHA in adapters with standard causal attention to ascertain whether the improvement mainly arises from an increased parameter count. From the last three rows, it is evident that CHA has a greater contribution ($+0.25$) than adding parameters ($+0.06$).

The two rightmost columns in Tab. 7 present the ablation study results on OOD benchmarks. We find that the three proposed components contribute variably across different benchmarks. Specifically, CHA consistently enhances generalization, while the other two components slightly reduce performance on TLP.

# 6 Conclusion

We have presented an AMR parser with new target forms of AMR graphs, explicit structure modeling with causal hierarchical attention, and the integration of pointer nets. We discussed and empirically compared multiple modeling options of CHA and the integration of CHA with the Transformer decoder. Eventually, CHAP outperforms all previous models on in-distribution benchmarks in the setting of no additional data, indicating the benefit of structure modeling.

## Limitations

We focus exclusively on training our models to predict AMR graphs using natural language sentences. Nevertheless, various studies suggest incorporating additional training objectives and strategies, such as BiBL and LeakDistill, to enhance performance. These methods also can be applied to our model.

There exist numerous other paradigms for semantic graph parsing, including Discourse Repre-

sentation Structures. In these tasks, prior research often employs the PENMAN notation as the target form. These methodologies could also potentially benefit from our innovative target form and structural modeling. Although we do not conduct experiments within these tasks, results garnered from a broader range of tasks could provide more compelling conclusions.

## Acknowledgments

We thank the anonymous reviewers for their constructive comments. This work was supported by the National Natural Science Foundation of China (61976139).

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

# A  More Plots

## A.1  Tree View of the Base Layer

Fig. 6 demonstrates the tree structure of the base layer. There are two main differences from the DAG representation of AMR: (1) Non-leaf nodes have no label. Instead, the label is a child node. (2) relations are presented as nodes instead of labeled edges.

## A.2  Attention Mask with Broken Structural Locality

Fig. 5 shows the two variants of breaking structural localities, which is discussed in Sec 5.2.

# B  Generate Attention Mask for Bottom-up Generation

Alg. 2 shows the procedure of generating $M_{\text{CHA}}$ for $\Uparrow$.

# C  Metrics

We report following fine-grained metrics:

- No WSD: This process computes while disregarding PropBank senses.

- Wikification: This denotes the F-score related to the Wikification task.

- Concepts: This signifies the F-score for the Concept Identification task.

- NER: This pertains to the F-score associated with the Named Entity Recognition task.

- Negations: This involves the F-score for the Negation Detection task.

- Unlabeled: This involves computations on the predicted graphs after all edge labels have been removed.

- Reentrancy: This is computed solely on reentrant edges.

- Semantic Role Labeling (SRL): This is computed only for ':ARG-i roles'.

# D  Implementation Details

We use the BART-base and BART-large checkpoints downloaded from the `transformer` library to initialize our models. An augmented vocabulary is used, which includes additional AMR relation

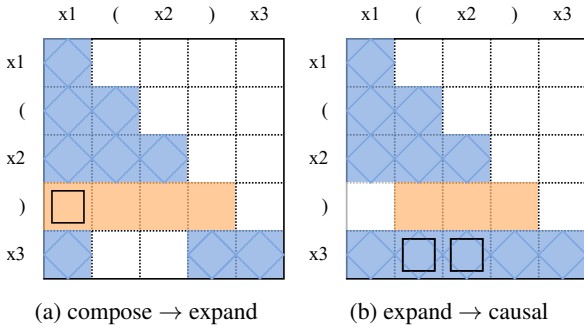

(a) compose → expand     (b) expand → causal

Figure 5: The target forms and the attention mask of two variants of ⇓single. Black squares mean that the cell is changed.

tokens, such as :arg0. Unlike other translation-based models, we do not add predicates (e.g., have-condition-91) into the vocabulary because they have ingorable effects on performance according to our preliminary experiments.

We train our models for 50,000 steps, using a batch size of 16. This amounts to approximately 22 epochs on AMR 2.0 and around 15 epochs on AMR 3.0. We use an AdamW optimizer (Loshchilov and Hutter, 2019), accompanied by a cosine learning rate scheduler (Loshchilov and Hutter, 2017) with a warm-up phase of 5,000 steps. The peak learning rate is set at $5 \times 10^{-5}$ for base models and $3 \times 10^{-5}$ for large models. We use one NVIDIA TITAN V to train models based on BART base, costing about 6 hours, and use one NVIDIA A40 to train models based on BART large, costing about 15 hours.

---

**Algorithm 2:** $M_{\mathrm{CHA}}$ for ⇑.

**Data:** sequence of token $t$ with length $N$,
       sequence of pointers $s$ with length $N$
**Result:** attention mask $M_{\mathrm{CHA}} \in \mathbb{R}^{N \times N}$
$S \leftarrow [\,]$              ▷ Empty stack
$M_{\mathrm{CHA}} \leftarrow -\infty$
**for** $i \leftarrow 1$ **to** $N$ **do**
    **if** $t[i] =' \blacksquare'$ **then**        ▷ compose
       $j \leftarrow i$
       **while** $j > s[i]$ **do**
          $M_{\mathrm{CHA}}[ij] \leftarrow 0$
          $j \leftarrow S.pop()$
       **end**
       $M_{\mathrm{CHA}}[ij] \leftarrow 0$
       $S.push(i)$
    **else**
       $S.push(i)$
       **for** $j \in S$ **do**        ▷ expand
          $M_{\mathrm{CHA}}[ij] \leftarrow 0$
       **end**
    **end**
**end**
**return** $M_{\mathrm{CHA}}$

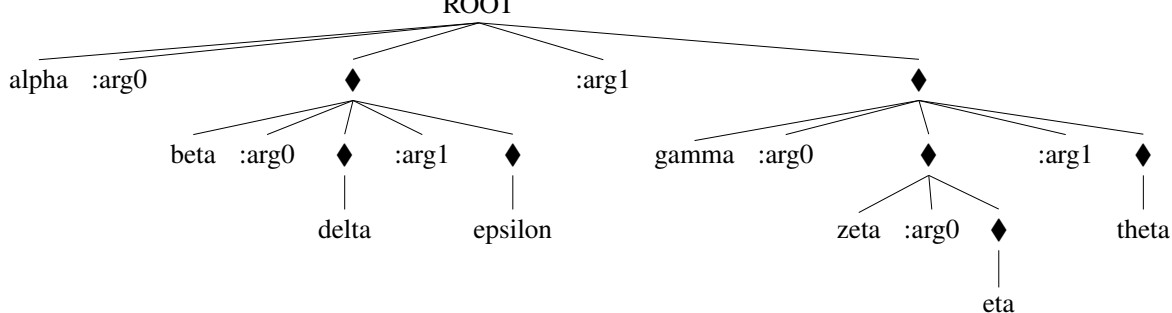

(a) The full tree in Fig. 2 without auxiliary tokens.

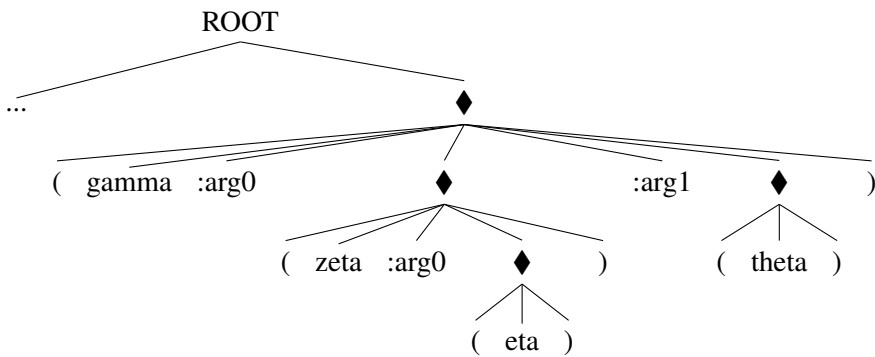

(b) The tree related to gamma in Fig. 2 in the ⇓single form.

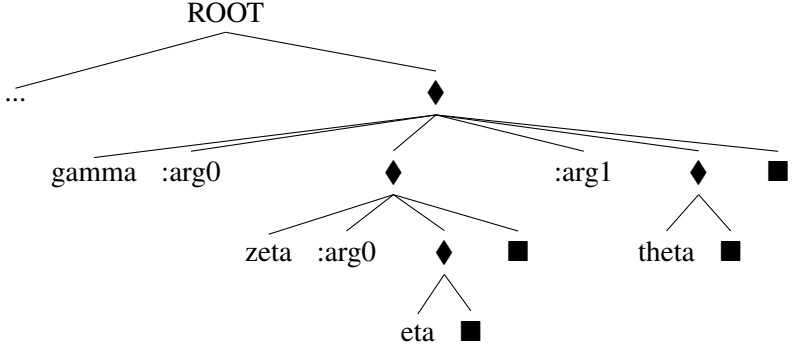

(c) The tree related to gamma in Fig. 2 in the ⇑ form.

Figure 6: The structure modeled in the base layer.