# OpenReview forum: "AMR Parsing with Causal Hierarchical Attention and Pointers"
_EMNLP/2023/Conference — EMNLP 2023 Main_

### Official Review · Reviewer_MHyS · 2023-08-01

**Typos Grammar Style And Presentation Improvements:** 1. The figure placement seems a littl…
**Soundness:** 4

**Excitement:**

3: Ambivalent: It has merits (e.g., it reports state-of-the-art results, the idea is nice), but there are key weaknesses (e.g., it describes incremental work), and it can significantly benefit from another round of revision. However, I won't object to accepting it if my co-reviewers champion it.

**Paper Topic And Main Contributions:**

The paper focuses on AMR genration using a neural translation architecture extended by a pointer mechanism. The authors contribute
* new AMR representations/linearizations that involve pointers to dismiss variables, and one that also uses pointers instead of parentheses to encode parent-child-relations
* an extension of the decoding process with hierarchical attention to enforce local contributions from child node encodings to parents

**Reasons To Accept:**

The contributed representations seem very sensible to me to eliminate variables, specifically the last one follows a trend to remove output representation complexity and modelling relations within the output explicitly. Their ablation study suggests that each single of their contributed additions to the parsing model did benefit the prediction accuracy.

**Reasons To Reject:**

I have no serious reasons to reject the paper, but stumbled upon some unclarities and issues in the presentation (listed below). I'd suggest they can be ironed out within an iteration to a camera-ready version.

**Reproducibility:**

4: Could mostly reproduce the results, but there may be some variation because of sample variance or minor variations in their interpretation of the protocol or method.

**Reviewer Confidence:**

3: Pretty sure, but there's a chance I missed something. Although I have a good feel for this area in general, I did not carefully check the paper's details, e.g., the math, experimental design, or novelty.

---

> ### Author Rebuttal · Authors · 2023-08-29
>
> Thank you for your constructive comments and suggestions, and they are exceedingly helpful for us to improve our paper.
>
> > 2\. Tab. 1 references a figure in the appendix and I don't understand why. Was this meant to ref. to Fig. 3?
>
> Sorry for the confusion. All references to Fig. 5 in the main text should reference Fig. 3 instead.
>
> > 5\. In Alg. 1: Something seems off.
>
> You are right. `S.push` should be performed in both cases. We will also fix the issue of iteration range.
>
> > 6\. Sec. 3.4 describes the target encoding $\Uparrow$
>  as bottom-up generation. I disagree, as all nodes are still introduced in top-down fashion, they are only completed bottom-up.
>
> The direction is defined with respect to the syntax tree of linearized AMR, instead of the AMR graph. In a syntax tree, all tokens in an AMR graph (e.g., `alpha` and `:arg0`) are leaves and their compositions are non-leaf nodes. In our bottom-up generation, we first generate tokens and then their compositions (represented by `■` and blue pointers in Tab 1.e). We compare syntax trees and AMR graphs in lines 223-236 and Appendix A.1. We will use a more precise presentation in our revised paper.
>
> The `top-down fashion` in your question actually refers to `the pre-order DFS` graph linearization of AMR graphes, which is orthogonal to the syntax tree generation direction defined above.
>
>
> > 7\. line 401: Which trained model do you use to obtain silver data?
>
> We train CHAP three times. Then, the silver data is obtained using the 2nd best model according to the metric on the validation set. We only generate the silver data once due to the high computational cost of predicting AMR graphs on a massive corpus.
>
> > 1\. figure placement;
> >
> > 3\. accessible visual effects;
> >
> > 4\. the figures illustrate different AMR graphs;
> >
> > 8\. entangled writing;
> >
> > 9\. too little explanation in the introduction;
> >
> > 10\. wrong pre- and surnames in the citations;
> >
> > 11\. typo.
>
> Thank you for bringing up these issues and we will revise our paper accordingly.

---

### Official Review · Reviewer_9uSY · 2023-08-02

**Soundness:** 4

**Excitement:**

4: Strong: This paper deepens the understanding of some phenomenon or lowers the barriers to an existing research direction.

**Paper Topic And Main Contributions:**

This paper introduces a new method for AMR parsing called CHAP. In contrast to recent work which has focused on using pretrained transformer decoders to output linearized AMR graphs, CHAP attempts to add graph structure to the decoder architecture, while still maintaining compatibility with pretrained transformer decoders (namely BART). The new decoder architecture has three major modifications: (1) a new hierarchical attention mechanism based on Transformer Grammars (Sartran et al. 2022) is used in some decoder layers; (2) the transformer decoder generates a linearized tree representing a depth-first traversal of the AMR ("base layer"); (3) a "coreference layer" then uses an attention-based pointer mechanism to connect nodes in the tree that refer to the same thing, forming a graph with reentrancies. A benefit of using pointers is that it eliminates arbitrarily-named variables from the output.

The main experimental results are:
1. They test multiple variants of TG-style hierarchical attention and find that the original one works best.
2. They test multiple ways of incorporating hierarchical attention into pretrained decoder layers and settle on an approach that uses it in parallel with the standard attention mechanism.
3. They show that CHAP performs better than several baselines on the AMR 2.0 and AMR 3.0 datasets, except for a concurrent model called LeakDistill, which uses additional alignment data.
4. They show that CHAP has better performance on 2 of 3 out-of-distribution benchmarks.
5. An ablation study showing that only the combination of all aspects of CHAP results in the best performance on AMR 3.0.

Overall, this paper proposes a novel architecture for AMR parsing and shows that it outperforms all recent baselines except for a concurrent project called LeakDistill, which uses some additional training data.


**Questions For The Authors:**

A. 425: How did you pick this value for $\alpha$?

B. 430: Could you elaborate a little on how these models work, and what category they fall under according to Table 2?

C. Tab 4, Tab 6: Some of these results are pretty close. Can you include std devs or significance testing?

D. 476: What about these datasets makes them out of distribution? Is it the size of the graphs? The subject matter? How do we know CHA's structure modeling is what's helping?

E. Can you describe how LeakDistill works a little? What about it makes it so much better than CHAP?

F. Because edges in a graph are unordered, a single AMR graph can have multiple ways of turning it into a tree using DFS. How do you pick which one to use? What is your algorithm for translating AMR graphs into trees? Do you foresee any generalization issues because of this?

G. In the big picture, it seems like the ideal decoder architecture for AMRs would be some sort of graph NN, rather than a tree + pointer mechanism. Is the main selling point of decomposing the decoder pipeline into tree and pointer phases that it makes it easier to use with a pretrained transformer decoder? Or would this be possible with a graph NN too?

H. I don't understand the distinction between the "pointer encoder" and "pointer net" mentioned at 86. Are there two different places where pointers are used (aside from the pointers used in bottom-up parsing)? Same question for points (1) and (3) at 313.


**Reasons To Accept:**

The paper is well-written, well-organized, and well-motivated, and for the most part it was enjoyable to read. Sec 2.1 nicely lays out the advantages of their method compared to prior work. As a result, I actually have less to say than usual in terms of praise and criticism. This paper would clearly be a good fit at EMNLP.

The method itself seems to be an innovative improvement over prior work. The authors compare against an excellent variety of recent baselines on multiple AMR parsing datasets. I don't see any serious issues with the experimental methodology.

It's also very nice that the authors explored multiple alternatives for CHA, justifying the architecture they eventually settled on.


**Reasons To Reject:**

My biggest complaint about this paper is that in the experimental section, a lot of the SMATCH scores are very close, and I would like to see std devs or significance testing. More than three runs might also bolster confidence in these scores. Without this, it's hard to have confidence in some of the experimental results.

478: It's not clear to me that CHA is the cause of better generalization, and I think this claim requires a lot more justification.


**Reproducibility:**

4: Could mostly reproduce the results, but there may be some variation because of sample variance or minor variations in their interpretation of the protocol or method.

**Reviewer Confidence:**

3: Pretty sure, but there's a chance I missed something. Although I have a good feel for this area in general, I did not carefully check the paper's details, e.g., the math, experimental design, or novelty.

**Typos Grammar Style And Presentation Improvements:**

A. Fig 1: The example sentence isn't grammatically correct. Shouldn't it be "The employee liked his (or her) city tour"?

B. Fig 2: I think it would be more helpful to use a sentence in plain English as the running example, or at least explain why it's not necessary to use a plain English example.

C. 111: Since factorization-based models are a whole column in Tab 2, could you spend a little time explaining what they are?

D. 151: researches -> research

E. 276: unaware -> unaware of

F. 408: What is a wiki tag?

G. 435: Structual -> Structural

H. Is higher better for Smatch? Adding an arrow by Smatch would help.

I. 464: worthy -> valuable

J. It might be helpful to remind the reader, early on in the paper, that the task of AMR parsing is to receive a sequence as input and produce an AMR graph as output. This would help explain why the changes are being made to the decoder.

K. Footnote 6: works -> work

L. The way Table 6 is organized makes it hard to read. Maybe group models based on training data. The shading makes it seem like the shaded results don't matter or aren't as important.

M. Is there a result missing from Tab 7?

---

> ### Author Rebuttal · Authors · 2023-08-29
>
> Thanks for your encouraging words and constructive comments. Our point-to-point responses to your comments are given below.
>
> > **Reason To Reject 1** Expect std devs or significance testing.
> >
> > **Question C. Tab 4, Tab 6** Some of these results are pretty close. Can you include std devs or significance testing?
>
> In general, our models have a low std dev. We will report std devs in our revised paper. We report some std devs below (new scores are bolded):
>
> Based on Tab. 4: The influence of different architectures.
>
> | Architecture | Smatch | Std dev  |
> | ------------ | :----: | :------: |
> | Parallel     | 82.63  | **0.02** |
> | Pipeline     | 82.59  | **0.05** |
> | Inplace      | 82.43  | **0.04** |
> | w/o CHA      | 82.38  | **0.12** |
>
> Based on Tab. 5: Fine-grained Smatch scores on in-domain benchmarks (trained on AMR 3.0 without silver data).
>
> | Model       |  Smatch   | Std dev  |
> | ----------- | :-------: | :------: |
> | Ancestor    |   83.5    |    -     |
> | BiBL        |   83.9    |    -     |
> | LeakDistill |   84.5    |    -     |
> | CHAP        | **84.37** | **0.04** |
>
> Based on Tab. 6: Fine-grained Smatch scores on in-domain benchmarks (trained on AMR 3.0 with silver data).
>
> | Model       | TLP: Smatch | TLP: Std dev | Bio: Smatch | Bio: Std dev |
> | ----------- | :---------: | :----------: | :---------: | :----------: |
> | LeakDistill |    82.6     |      -       |    64.5     |      -       |
> | CHAP        |    82.7     |   **0.14**   |    66.1     |   **0.46**   |
>
>
> > **Reason To Reject 2** The claim that CHA is the cause of better generalization requires a lot more justification.
> >
> > **Question D. 476** What about these datasets make them out of distribution? How do we know CHA's structure modeling is what's helping?
>
> The texts in these datasets are of different subject matters. However, these datasets share the same label set and annotation guidelines.
>
> We agree that the claim at line 478 should be justified because the reported results only show CHAP's generalization ability. Therefore, we evaluate the gains from each component (i.e., CHA, pointer net, pointer encoder) and report the results below for the setting of Tab. 7 Ablation Study. New scores are bolded. Although models with these components outperform the baseline, we do not find a clear cause or consistent trend in the results. Therefore, we will claim CHAP rather than CHA helps generalization in our revised paper.
>
> | Encoding pointers | Adapter | Coref. layer | TLP      | Bio      |            |
> | ----------------- | ------- | ------------ | -------- | -------- | ---------- |
> | yes               | CHA     | yes          | 81.8     | 65.1     |            |
> | no                | CHA     | yes          | **81.7** | **64.8** |            |
> | no                | CHA     | no           | **82.1** | **64.6** |            |
> | no                | no      | no           | **81.6** | **64.3** | (baseline) |
>
>
> > **Question A. 425** How did you pick this value for $\alpha$?
>
> We run a grid search over the set $\{0.01, 0.05, 0.075, 0.1, 1\}$ and choose the value resulting in the best validation performance.
>
> > **Question B. 430** Could you elaborate a little on how these models work, and what category they fall under according to Table 2?
> >
> > **Question E** Can you describe how LeakDistill works? What about it makes it so much better than CHAP?
>
> All these models are translation-based models. We will explain them in the revised paper. Here is a brief introduction:
>
> 1. SPRING is a finetuned BART.
> 2. Ancestor enhances the decoder of SPRING with ancestors in AMR graphs.
> 3. BiBL and AMRBART enhance SPRING with auxiliary training losses.
> 4. LeakDistill trains a SPRING with leaked information and then distills it to a standard SPRING. We have discussed the success of LeakDistill at lines 460-464.
>
> Transition-based and factorization-based models have weaker performance, so we do not list their scores in our paper.
>
> > **Question F** Because edges in a graph are unordered, a single AMR graph can have multiple ways of turning it into a tree using DFS. How do you pick which one to use? What is your algorithm for translating AMR graphs into trees? Do you foresee any generalization issues because of this?
>
> Following previous work, we use the DFS order provided in the AMR datasets. We believe that the DFS algorithm does not raise a generalization issue:
>
> * Regarding training, previous work[1] suggests that "any depth first traversal of the AMR is an effective linearization."
> * Regarding evaluation, the Smatch metric is invariant to different DFS algorithms.
>
> [1] Neural AMR: Sequence-to-Sequence Models for Parsing and Generation (Konstas et al., ACL 2017)
>
> > **Question G** It seems like the ideal decoder architecture for AMRs would be some sort of graph NN, rather than a tree + pointer mechanism. Is the main selling point of decomposing the decoder pipeline into tree and pointer phases that it makes it easier to use with a pretrained transformer decoder? Or would this be possible with a graph NN too?
>
> Previous translation-based models (e.g., SPRING) can be easily used with a pretrained decoder by linearizing AMR graphs to texts. Our decomposition aims to use a pretrained decoder better (discussed in Sec 3 lines 161-164 and 171-174).
>
> The decoding procedure involves a growing graph, making the application of GNNs non-trivial. We are open to the idea of utilizing GNNs and expect an exploration in future work.
>
> > **Question H** I don't understand the distinction between the "pointer encoder" and "pointer net" mentioned at 86. Are there two different places where pointers are used (aside from the pointers used in bottom-up parsing)? Same question for points (1) and (3) at 313.
>
> "Pointer net" (or "pointer mechanism"), introduced in Sec 4.3, is used to predict pointers. In contrast, a "pointer encoder", introduced in Sec 4.1, encodes previously predicted pointers (just like the word embedding layer in BART's decoder encoding previously predicted tokens).
>
> > **Typos ... C. 111** Since factorization-based models are a whole column in Tab 2, could you spend a little time explaining what they are?
>
> Factorization-based models factorize the score of a graph into parts for smaller subgraphs. They operate graphs directly, so their decoding is generally much more complicated. For example, [2] predicts a subgraph for each token and then composes them to obtain the final AMR graph. [3] predicts an AMR graph by incrementally and iteratively predicting new edges and nodes.
>
> [2] AMR dependency parsing with a typed semantic algebra (Groschwitz et al., ACL 2018)
>
> [3] AMR Parsing via Graph-Sequence Iterative Inference (Cai & Lam, ACL 2020)
>
>
> > **Typos ... F. 408** What is a wiki tag?
>
> All named entities in AMR graphs are linked to a Wikipedia page and have a wiki tag containing the page name.
>
> > **Typos ... H** Is higher bettter for Smatch?
>
> Yes.
>
> > **Typos .. M** Is there a result missing from Tab 7?
>
> Yes, the missing value is `83.79`.

---

### Official Review · Reviewer_2oGz · 2023-08-03

**Soundness:** 3

**Excitement:**

3: Ambivalent: It has merits (e.g., it reports state-of-the-art results, the idea is nice), but there are key weaknesses (e.g., it describes incremental work), and it can significantly benefit from another round of revision. However, I won't object to accepting it if my co-reviewers champion it.

**Paper Topic And Main Contributions:**

This paper presents a novel AMR parser. The authors present to augment a translation-based approach with a decoder that can model structure. The goal is to be able to benefit from pre-trained models while exploiting the structure in the output space.
The paper is well written and clear and the authors show that their approach matches the recently proposed state-of-the-art LeakDistill approach in terms of accuracy.

**Reasons To Accept:**

- Sound and clean approach.
- State-of-the-art results.

**Reasons To Reject:**

- Somewhat narrow applicability. No other applications beyond AMR parsing are discussed.

**Reproducibility:**

4: Could mostly reproduce the results, but there may be some variation because of sample variance or minor variations in their interpretation of the protocol or method.

**Reviewer Confidence:**

3: Pretty sure, but there's a chance I missed something. Although I have a good feel for this area in general, I did not carefully check the paper's details, e.g., the math, experimental design, or novelty.

---

> ### Author Rebuttal · Authors · 2023-08-29
>
> Thanks for your review and we appreciate your time in reading the paper.
>
> > **Reason To Reject 1** Somewhat narrow applicability. No other applications beyond AMR parsing are discussed.
>
> We have discussed this in the Limitation section. However, we would like to point out that improving performance on one task is a common type of contributions in \*ACL conferences and the ACL23 review policy explicitly argues against regarding this as a weakness:
>
>     "A main track paper may well make a big contribution to a narrow subfield."
>
> We hope you could reconsider your evaluation based on the review policy.

---

### Meta-Review · Area_Chair_n6pf · 2023-09-22

**Recommendation:** 4

**Metareview:**

As reviewers indicate, this paper introduces a new method for AMR parsing called CHAP. In contrast to recent work which has focused on using pretrained transformer decoders to output linearized AMR graphs, CHAP attempts to add graph structure to the decoder architecture, while still maintaining compatibility with pretrained transformer decoders (namely BART). The new decoder architecture has three major modifications: (1) a new hierarchical attention mechanism based on Transformer Grammars (Sartran et al. 2022) is used in some decoder layers; (2) the transformer decoder generates a linearized tree representing a depth-first traversal of the AMR ("base layer"); (3) a "coreference layer" then uses an attention-based pointer mechanism to connect nodes in the tree that refer to the same thing, forming a graph with reentrancies. A benefit of using pointers is that it eliminates arbitrarily-named variables from the output.

The method itself seems to be an innovative improvement over prior work.The main experimental results pointed out by reviewers are the following:

    They test multiple variants of TG-style hierarchical attention and find that the original one works best.
    They test multiple ways of incorporating hierarchical attention into pretrained decoder layers and settle on an approach that uses it in parallel with the standard attention mechanism.
    They show that CHAP performs better than several baselines on the AMR 2.0 and AMR 3.0 datasets, except for a concurrent model called LeakDistill, which uses additional alignment data. The authors explored multiple alternatives for CHA, justifying the architecture they eventually settled on.
    They show that CHAP has better performance on 2 of 3 out-of-distribution benchmarks.
    An ablation study showing that only the combination of all aspects of CHAP results in the best performance on AMR 3.0.

Additionally, the  paper is well written and clear, well-organized, and well-motivated.

According to reviewers there are not many reasons to reject the paper.

    A reviewer indicates that in the experimental section, a lot of the SMATCH scores are very close, and  std devs or significance testing is recommended. The authors addressed this in the rebuttal.
    Somewhat narrow applicability. No other applications beyond AMR parsing are discussed.
    There are some unclarities an issues pointed out by Reviewer 3. Tthey can be ironed out within an iteration to a camera-ready version.

All in all, it seems that the authors could address the shortcomings in the final version of the paper.

---

### Decision · Program_Chairs · 2023-10-07

**Decision:**

Accept-Main

**Comment:**

As reviewers indicate, this paper introduces a new method for AMR parsing called CHAP. In contrast to recent work which has focused on using pretrained transformer decoders to output linearized AMR graphs, CHAP attempts to add graph structure to the decoder architecture, while still maintaining compatibility with pretrained transformer decoders (namely BART). The new decoder architecture has three major modifications: (1) a new hierarchical attention mechanism based on Transformer Grammars (Sartran et al. 2022) is used in some decoder layers; (2) the transformer decoder generates a linearized tree representing a depth-first traversal of the AMR ("base layer"); (3) a "coreference layer" then uses an attention-based pointer mechanism to connect nodes in the tree that refer to the same thing, forming a graph with reentrancies. A benefit of using pointers is that it eliminates arbitrarily-named variables from the output.

The method itself seems to be an innovative improvement over prior work.The main experimental results pointed out by reviewers are the following:

    They test multiple variants of TG-style hierarchical attention and find that the original one works best.
    They test multiple ways of incorporating hierarchical attention into pretrained decoder layers and settle on an approach that uses it in parallel with the standard attention mechanism.
    They show that CHAP performs better than several baselines on the AMR 2.0 and AMR 3.0 datasets, except for a concurrent model called LeakDistill, which uses additional alignment data. The authors explored multiple alternatives for CHA, justifying the architecture they eventually settled on.
    They show that CHAP has better performance on 2 of 3 out-of-distribution benchmarks.
    An ablation study showing that only the combination of all aspects of CHAP results in the best performance on AMR 3.0.

Additionally, the  paper is well written and clear, well-organized, and well-motivated.

According to reviewers there are not many reasons to reject the paper.

    A reviewer indicates that in the experimental section, a lot of the SMATCH scores are very close, and  std devs or significance testing is recommended. The authors addressed this in the rebuttal.
    Somewhat narrow applicability. No other applications beyond AMR parsing are discussed.
    There are some unclarities an issues pointed out by Reviewer 3. Tthey can be ironed out within an iteration to a camera-ready version.

All in all, it seems that the authors could address the shortcomings in the final version of the paper.